# Assessment of the Health Care System in Poland and Other OECD Countries Using the Hellwig Method

**DOI:** 10.3390/ijerph192416733

**Published:** 2022-12-13

**Authors:** Daria Smarżewska, Wioletta Sylwia Wereda, Joanna Anna Jończyk

**Affiliations:** 1Institute of Management and Quality Science, Faculty of Engineering Management, Bialystok University of Technology Kleosin, 16-001 Bialystok, Poland; 2Faculty of Security, Logistics and Management, Warsaw Institute of Organization and Management, Military University of Technology, 00-908 Warsaw, Poland

**Keywords:** health care system, health care systems ranking, polish health care system, indicators of health care systems

## Abstract

The health care system is a key element in the functioning of any country. However, depending on the level of funding, the number of medical staff and their availability, there are significant discrepancies in the health care systems of different countries. This article presents a picture of the Polish health care system compared to the systems of other selected OECD countries. The comparison was made on the basis of selected indicators concerning financing, medical and nursing staff as well as patient satisfaction with the availability and quality of healthcare. The aim of this article is to analyze the Polish health care system and compare it with other selected OECD countries’ health care systems. A literature review, secondary data analysis and statistical analyses were used as the research method. The ranking was prepared using Hellwig’s linear ordering method. Ten indicators related to financing, medical and nursing staff, and residents’ opinions on the availability and quality of medical services were selected for the analysis. The presented results clearly indicated that Norway, Germany and Switzerland have the best health care systems. The Polish system takes one of the last places in developed classification. The conducted analysis indicates the need to introduce changes to the health care system and the need to implement solutions from countries where health care systems have been indicated as the best.

## 1. Introduction

The World Health Organization defines the health care system as “all organizations, expenditures and institutions which assumption is to generate activities aimed at improving health” [1,2]. The Polish health care system operates on the basis of a model based on universal and compulsory health insurance [3]. This means that funds allocated to financing services may come from voluntary funds, i.e., private patients’ funds or compulsory ones, paid as public health insurance contributions [4]. Adequate protection of the health condition of citizens is related to the achievement of the macroeconomic goals of the health care system. These include the level of health costs and increasing the effectiveness of healthcare entities [5,6]. The efficient functioning of the health care system translates into the economic situation of the country. In crisis situations, such as the COVID-19 pandemic, all errors in the system quickly become apparent [7]. Michalik-Marcinkowska and Izdebski indicated that the main disadvantages of the Polish health system include an inadequate level and allocation of funds and staff shortages [8,9]. The analysis of ten key variables in selected OECD countries using the Hellwig’s linear ordering method allowed for the development of conclusions and an indication of the directions of changes in the Polish health care system [10].

The basic problem in the Polish health care system is its financing [11]. The adoption of various sources to finance the system resulted from the reforms and systemic changes that have taken place in Poland over the years [6]. The main sources of financing the system include health insurance contributions and the state budget. The amount of the health insurance contribution is set by law and amounts to 9% of the basis of its assessment [12]. The second source of fundraising for the system is the state budget. Funding from the budget covers mainly the State Emergency Medical Services as well as specialized services or activities aimed at restructuring, creating or relieving the debt of existing medical entities [12]. Depending on the entity creating the funds, the funds may also come from the state treasury budget or the budgets of local governments [12]. The organizer of the health care system in Poland is the government, acting through the Ministry of Health. The main tasks of the Ministry of Health include planning and shaping health policies [13]. The role of the payer in the system is performed by the National Health Fund. It consists of the headquarters and 16 voivodship branches [14]. The National Health Fund deals with the financing of health services provided as part of medical activities. The strategic role of the payer (NHF) in the process of resource distribution in healthcare entities including human resources is also analyzed in the context of other health care systems [15,16]. The funds at the disposal of the National Health Fund are collected from health insurance contributions [17]. 

Service providers are another key element of the health care system. These are all entities that perform medical activities. Pursuant to the Act of 15 April 2011 on medical activity, they are defined as “medical entities referred to in Art. 4, as well as doctors, nurses or physiotherapists practicing in the medical profession as part of medical practice as apprenticeships referred to in Article 5” [18]. In other words, the service provider can be both a hospital, a clinic, an ambulance station, as well as a person performing a medical profession in the form of an independent practice [19]. 

The last element in the Polish health care system is the recipient, i.e., the patient. The right to use health services financed from public funds is granted to people who are covered by both compulsory and voluntary health insurance [20]. The right to benefits is also granted to people who are under 18 years of age, provided that they have Polish citizenship or have obtained refugee status or subsidiary protection [20]. It should be noted that this right is also ensured by Art. 68 of the Polish Constitution, which states that every citizen has the right to health protection [21]. Pursuant to the Act on financing health services from public funds, “beneficiaries have the right to use health care services aimed at maintaining health, preventing diseases and injuries, early detection of diseases, treatment, care as well as preventing and limiting disability” [20]. 

The level of indebtedness of the health care system in Poland is significant. Achieving negative financial results by hospitals and the constant increase in their debt is one of the most important problems of the system [5]. Seventy percent (70%) of all the arrears in medical facilities concern payments to suppliers of drugs, materials and energy [22]. This situation is worrying because it has been going on for nearly 20 years. Attempts to introduce reforms aimed at reducing the level of indebtedness have not always turned out to be effective. The personnel situation in the Polish health care system is also worrying. The number of doctors per 1000 inhabitants in Poland is one of the lowest in Europe. The age of working doctors is also a negative factor—a large percentage are people over 50. The number of nurses in 2019 in Poland was 8.83 nurses/1000 inhabitants [23]. The largest group in this profession are people over 50 (over 60% of all employed persons) [24]. A current and important problem in the medical profession is the so-called double practice. It applies to both doctors and nurses. According to Socha and Bech [25], in the case of doctors, the cause of double practice is financial reasons. The same reasons are indicated for double practices among nurses. The lack of adequate financial resources, lack of employees and inadequate allocation of these resources are some of the problems that the Polish system has to face [26,27]. The article also uses the conclusions of the project financed by the National Research Center on the basis of Decision Project No. DEC-2011/03/B/HS4/04544.

It should be emphasized that the indicated implications are key and require careful analysis and assessment, especially in the conditions of the systematic increase in patients’ expectations regarding the quality and availability of health care [28,29].

## 2. Methods

The aim of this article is to analyze the Polish health care system and compare it with other selected OECD countries’ health care systems. To achieve this set goal, the following hypotheses were established:

**H1.** 
*The assessment of the functioning of health care systems is based on indicators related to finances, human resources and the availability and quality of medical services.*


**H2.** 
*A higher place in the ranking of health care systems means a higher evaluation of the health care system.*


**H3.** 
*Poland probably occupies one of the last places in the rankings.*


The analysis was carried out using data from the OECD Health at Glance 2021 report [23]. The basis for the selection of the variables was the analysis of the literature on the subject, which indicated the areas where there are problems in the health care systems. The following variables were selected for the preparation of the ranking:

X1—number of doctors per 1000 inhabitants;

X2—number of nurses per 1000 inhabitants;

X3—number of medical school graduates (doctors) per 100,000 inhabitants;

X4—nursing graduates per 100,000 inhabitants;

X5—health care expenditure as % of GDP;

X6—government spending on health care per capita (USD PPP);

X7—out-of-pocket expenditure on health care per capita (USD PPP);

X8—number of beds per 1000 inhabitants;

X9—% of the population covered by primary health care;

X10—% of the population satisfied with the availability and quality of healthcare.

In order to create the ranking, the Hellwig linear ordering procedure was carried out, which is one of the standard ordering methods. The following construction of the synthetic measure of Hellwig was used [10,30]

Normalization of variables:

(1)$$z_{\mathrm{ij}}=\frac{\bar{x}\mathrm{ij}}{s_{j}}$$
where

$x_{\mathrm{ij}}$—observation of the j-th variable for the object i;

${\bar{x}}_{\mathrm{ij}}$—arithmetic mean of observations of the j-th variable;

$s_{j}$—standard deviation of observation of the j-th variable.

2.Pattern coordinates:



(2)
$$z_{\mathrm{ij}\;}=\{\begin{array}{c} {\max\;\{z}_{\mathrm{ij}}\}\;\mathrm{for}\;\mathrm{stimulant}\;\mathrm{variables}\\ {\max\;\{z}_{\mathrm{ij}}\}\;\mathrm{for}\;\mathrm{destimulant}\;\mathrm{variables} \end{array}$$



3.Object distances from the pattern:



(3)
$$d_{i0}=\sqrt{\overset{m}{\underset{j=1}{\sum }}({\;z}_{\mathrm{ij}}-z_{0j}})$$



4.Aggregate variable values:

(4)$$d_{i\;}=1-\frac{d_{i0}}{d_{0}}$$
where



(5)
$$d_{0\;}=\bar{d_{0\;}}+2s_{d}$$





(6)
$$\bar{d_{0}\;}=\frac{1}{n}\;\sum^{\;}\begin{array}{c} n\\ i=1 \end{array}d_{i0}$$





(7)
$$s_{d}=\sqrt{\sum^{\;}\begin{array}{c} n\\ i=1\; \end{array}\left(d_{i0}-\bar{d_{0}}\right)\begin{array}{c} 2 \end{array}\;}$$



It was assumed that max {q_i_} was the best object while min {q_i_} was the worst. It was also assumed that all the factors taken into account while creating the ranking, except for the out-of-pocket expenditure for health protection per capita, were stimulants. The values of the synthetic measure in the Hellwig method are usually in the range [0, 1] [10]. For the purposes of the conducted analysis, equal weight values—1—were assumed for all the analyzed factors. The analysis of linear ordering made it possible to create a ranking for the selected European countries.

## 3. Results

Healthcare systems vary from country to country. These differences mainly concern the method of financing health services. There are two basic forms of financing the system: public funds (e.g., taxes, compulsory insurance) and private funds (the so-called out-of-pocket expenses) [4]. In addition to financial conditions, the shape of the systems is also influenced by political, economic and historical factors [1]. One of the most popular forms of comparing the level of expenditure on health care is to indicate it as the % of the GDP of a given country (Figure 1).

In 2019, the average percent of the GDP allocated to financing health care was 8.86% (the average was calculated on the basis of the data for all the analyzed countries). In 17 countries, this indicator was below the calculated average (Figure 1). The country that spent the highest percentage of GDP on health expenditures was the United States, and the lowest was Turkey. The level of GDP allocated to health care in Poland was low compared to other countries. 

An important indicator of the functioning of health systems is expenditure. In the analyzed countries, the average per capita government spending on health was USD 3170.12 (Table 1). Again, the United States ranked first for health expenditure per capita, followed by Mexico. The situation changed in the case of the level of the residents’ own expenses. Switzerland was the country where citizens spent the most private resources on health, and Turkey the least. In Poland, the government spent an average of USD 1643 per capita. In this country, the citizens bore their own medical expenses at the level of USD 646 per year. The high level of the citizens’ own expenditure on health care may have resulted from the lack of financing of some services [31]. The waiting time for visits was also a problem. Along with the increase in the demand for health services, their supply did not increase, which in turn causes queues to hospitals, clinics or specialists [31].

It should be noted that healthcare professionals are a key element in the functioning of this system [1]. The employment rate in the system accounted for over 10% of total employment in some OECD countries [32]. Despite the constant development of the health sector, many countries still have low or insufficient human resources [33]. The number of medical university graduates in the analyzed countries varied. The countries where the most people graduated from medical studies were Lithuania, Latvia, Denmark and Ireland (Table 2). The reverse was true for Luxembourg, Poland and Hungary. In the case of nursing graduates, the highest number of graduates were in Switzerland (over 100 people/100,000 inhabitants) and Norway (Table 2). The fewest people chose this profession in Luxembourg, Spain and Slovakia. Again, the analyzed indicators were low in the case of Poland (Table 2). Encouraging young people to choose medical professions is an important task for every country. It is estimated that by 2030 there will be a shortage of 3.43 million doctors and 8.22 million nurses in Europe [34].

One of the main problems in the Polish health care system is shortages of medical staff [1]. Compared to other countries, the number of doctors and nurses per 1000 inhabitants in Poland is one of the lowest. Countries such as Norway and Germany are distinguished by a high level of employment in these professional groups. It should be noted that healthcare professionals are a key element in the functioning of this system. The employment rate in the system accounts for over 10% of total employment in some OECD countries [31]. Despite the constant development of the health sector, many countries still have low or insufficient human resources [33]. The number of medical university graduates in the analyzed countries varied (Table 2). The countries where the most people graduated from medical studies were Lithuania, Latvia, Denmark and Ireland (Table 2). The reverse was true for Israel, Japan and Korea. In the case of nursing graduates, the most people graduated from these studies in Korea, Switzerland and Australia (over 100 people/100,000 inhabitants). The fewest people chose this profession in Mexico, Luxembourg and Colombia (Table 2). Again, the analyzed indicators were low in the case of Poland. 

An important element of the analysis and evaluation of health care systems is its availability. In most of the analyzed countries, between 98% and 100% of the population was covered by public health care (Table 3). Only in Mexico and the United States was the percentage of people in primary care less than 90%. The OECD data showed that in Poland, 93.4% of citizens had access to free, universal benefits, and only 26% of the total were satisfied with their quality and availability (Table 3). The countries with the highest scores for the health care system were Switzerland, Norway, the Netherlands and Belgium—in each of them, the proportion of satisfied patients was over 90%.

The analysis of linear ordering made it possible to create a ranking for the selected OECD countries. The data were standardized first (Table A1). After the standardization of the data, the values of the coordinates of the pattern were calculated (Table 4).

Next, the distance values were calculated from the pattern (Table 5). 

The obtained data were analyzed using the Hellwig correction. The ranking of health care systems is presented in Table 6. The countries were classified by the value of the synthetic measure of development (SMR_i_).

The presented ranking showed that in terms of the analyzed factors, the health care systems in Norway, Germany and Switzerland were in the forefront (Table 6). The last three places belonged to Chile, Colombia and Mexico. Poland was in 34th place. These results clearly indicate that the health care system in this country requires measures aimed at improving and increasing the analyzed indicators.

## 4. Discussion

The health care system is one of the most important elements of the functioning of a state. The level and forms of financing the system directly translate into the quality and availability of medical services. The analysis showed that the best health care system in terms of the analyzed factors was in Norway. In this country, the health benefits were 85%. Norway’s high level of GDP per capita expenditure kept the level of private expenditure on benefits low. The private health expenditure of residents relates mainly to the purchase of drugs, dental services or long-term care [35]. The health system in Norway is based on two types of service provision: specialist and basic. The first is financed by the government, while municipalities are responsible for the organization and financing of primary health care [36]. The Norwegian health care system is also focused on the development of long-term care, which is necessary due to the aging of the population [35]. An important action implemented by the Norwegian government is the constant reform of the system and adapting it to the current needs of society. Regarding Germany, the high level of access to medical services and qualified medical staff makes the German health care system one of the best in the world [37]. For example, in Germany, the health insurance contribution is 14.6% of a person’s income [37]. The financing of the system in this country has various sources, with insurance premiums constituting the largest percentage [37]. In addition, this country has an extensive network of hospitals, covering every region of the country, which are equipped with the highest quality equipment. Switzerland is another country that occupies a leading position in the ranking of health care systems. The system in this country is based on a mixed model. Compulsory health insurance premiums are the main source of financing. In addition, every citizen is obliged to bear additional costs related to the use of benefits, the so-called copayment [38]. Switzerland also has significant human resources for doctors and nurses, which additionally affects the effectiveness of the system.

Warzecha presented a ranking of health care systems, in which she took into account 24 variables, including, e.g., the number of nurses and doctors and health care expenditure as a percent of GDP. The ranking that she prepared showed that Sweden, Finland, the Netherlands, Ireland and Spain were among the countries with the best health care systems in 2017 [39]. The differences between the results were mainly due to the adopted variables. In addition to economic factors, Warzecha also took factors related to disease incidence, life expectancy, lifestyle and mortality into account—in this ranking, Poland took 22nd/28th place [39]. The cited studies and the ranking prepared in this publication indicate that despite attempts to improve the functioning of the health care system in Poland, it still differs significantly from the systems of other countries. 

An important issue in analyzing the health care system is the problem of human resources [31]. However, this problem does not only concern the number of system employees. System decision makers should focus not only on encouraging people to work in medical professions, as it is important to also properly understand the needs of staff depending on their workplace. The differences in the employment rate of health care workers in Europe are significant [31]. Kuhlumann et al. call for the development of a research program aimed at developing a methodology aimed at creating a health care system, the functioning of which will be focused on human resources [31]. The aforementioned proposal seems to be justified. Targeting employees of the health care system, learning about their needs, and researching and increasing their competences can significantly improve the quality of the services offered. System decision makers cannot make the right decisions without having specific knowledge. The number of doctors per 1000 inhabitants in more than half of the surveyed countries was less than four people. In more than half of the analyzed countries, the number of nurses did not exceed 10/1000 inhabitants. These data show that securing the health needs of citizens is significantly more difficult. Filling job gaps, including those related to the age of employees, should be a priority in the undertaken reforms to the system.

The situation of Poland compared to the leading countries shows that actions should be taken to improve the functioning of the health care system. According to Figure [38], in the Polish health care system, one should work on increasing the human resources of the system, changing the financing system (e.g., introducing copayments) or modernizing the guaranteed benefits package. In addition, the use of practices such as the decentralization of system financing, increasing health insurance contributions or the introduction of copayments could improve the functioning of the health care system.

## 5. Conclusions

To conclude, the health care system in each country should be adapted to meet the current health needs of a given society. The financing of the system should come from many sources, both in the form of public and private expenditure. Important elements of the system are hospital infrastructure (mainly the number of beds) as well as the number and availability of medical personnel. The proper operation of the system requires action in many areas and the cooperation of various institutions. Broad and effective campaigns should also be conducted to encourage people to choose a medical profession. The estimated level of staff shortages is one of the most important challenges faced by system decision makers. Therefore, it is necessary to search for good practices and implement them in the Polish health care system, following the example of countries with the best systems. The prepared ranking of health care systems provides the basis for the selection of countries from which the solutions used in the best of them could be implemented. The direction of further research should be a detailed comparative analysis of the Polish health care system with the countries with the best systems. 

The presented analysis made it possible to achieve the aim of the work and to verify our hypotheses:

H1: The assessment of the functioning of health care systems is based on indicators related to finances, human resources and the availability and quality of medical services—the indicated factors are key elements of the health care system. Their value determines the effectiveness of the functioning of the system and, consequently, the level of securing citizens with health services.

H2: A higher place in the ranking of health care systems means a higher evaluation of the health care system—examples of the highest rated systems are those in Norway, Germany and Switzerland, as these countries not only have a high level of financing for their systems, but they also have adequate staffing and provide relatively good access and a high quality of services.

H3: Poland probably occupies one of the last places in the ranking—Poland took 34th place in the ranking out of 37 possible spots.

The authors are aware of the limitations resulting from the conducted analyses. First of all, the number of factors selected for the analysis and the number of countries selected for the ranking can be indicated here. Due to the topicality and seriousness of the problem, research on the effectiveness of health care systems should be continued. The continuation of this type of analysis may allow for the construction of benchmark systems, enabling the comparison and improvement of systems in countries with low indicators of health care system functioning, an example of which is Poland.

## Figures and Tables

**Figure 1 ijerph-19-16733-f001:**
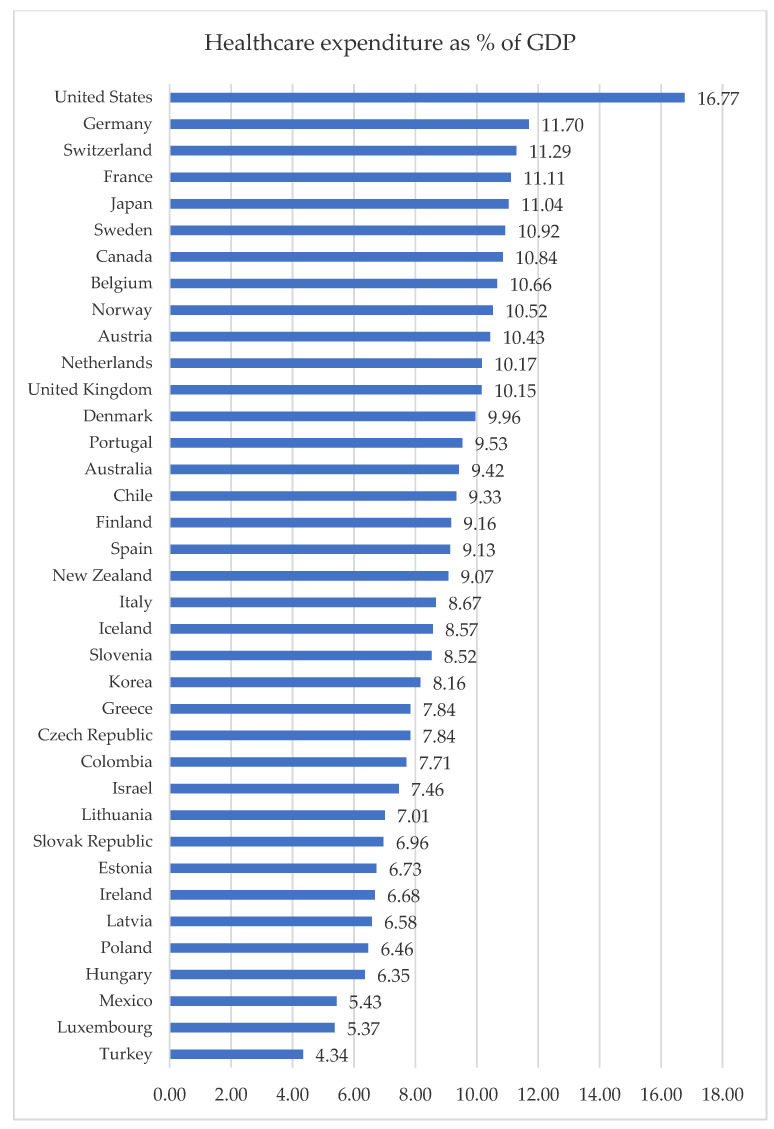
Healthcare expenditure as % of GDP in 2019 [23].

**Table 1 ijerph-19-16733-t001:** Health care expenditure by type in selected OECD countries in 2019 [23].

Country	Government Expenditure on Health Care per Capita (USD PPP)	Out-of-Pocket Expenditure on Health Care per Capita (USD PPP)
Mexico	558.64	574.33
Turkey	987.18	279.76
Columbia	989.43	286.78
Latvia	1275.18	798.82
Grecee	1385.76	930.01
Chile	1388.67	902.80
Hungady	1482.51	687.26
Poland	1643.20	646.11
Slovakia	1746.72	442.33
Lithuania	1810.38	916.19
Estonia	1867.59	639.48
Israel	1880.82	962.09
Portugal	2041.14	1306.29
Korea	2077.16	1329.09
Slovenia	2404.42	899.05
Spain	2542.50	1057.78
Italy	2700.57	952.83
Czech Republic	2795.55	621.94
New Zeland	3355.18	856.67
Australia	3378.81	1540.43
United Kingdom	3533.21	966.93
Finland	3550.34	1011.13
Iceland	3763.78	776.98
Canada	3768.30	1602.14
Irleand	3790.96	1292.25
Japan	3936.64	754.82
Belgium	4192.44	1265.96
Austria	4292.13	1412.97
France	4414.98	859.28
Denmark	4562.25	915.32
Luxemburg	4600.02	742.36
Sweden	4712.74	839.19
Netherlands	4742.98	996.22
Switzerland	4765.67	2372.39
Germany	5514.43	1003.57
Norway	5788.24	956.38
United States	9053.84	1894.63

**Table 2 ijerph-19-16733-t002:** Structure of doctors and nurses in selected OECD countries in 2019 [23].

Country	Doctors/1000 Inhabitants	Nurses/1000 Inhabitants	Number of Medical School Graduates (Doctors) per 100,000 Inhabitants	Nursing Graduates per 100,000 Inhabitants
Turkey	2.0	2.40	13.1	18.7
Colombia	2.3	1.39	12.5	7.7
Mexico	2.4	2.85	12.3	15.5
Poland	2.4	5.10	10.6	23.9
Korea	2.5	7.94	7.4	100.2
Japan	2.5	11.76	7.1	52.3
Chile	2.6	2.87	9.1	31.1
United States	2.6	11.98	8.1	65.6
Canada	2.7	9.98	7.6	52.7
United Kingdom	3.0	8.20	13.1	30.9
Luxembourg	3.0	11.72	0.00	10.7
Belgium	3.2	11.07	17.6	31.1
France	3.2	11.07	9.5	40.4
Finland	3.2	14.26	11.9	81.8
Latvia	3.3	4.39	23.5	26.9
Israel	3.3	5.01	7.2	26.6
Slovenia	3.3	10.28	13.8	78.1
Ireland	3.3	12.88	24.8	28.9
New Zealand	3.4	10.24	9.9	39.9
Estonia	3.5	6.24	10.4	28.9
Hungary	3.5	6.62	15.8	50.0
Slovak Republic	3.6	5.74	17.2	21.7
Netherlands	3.7	10.69	15.1	58.6
Australia	3.8	12.22	15.9	108.9
Iceland	3.9	15.36	11.4	59.6
Italy	4.1	6.16	17.6	18.4
Czech Republic	4.1	8.56	16.1	28.7
Denmark	4.2	10.10	23.0	44.7
Sweden	4.3	10.85	13.5	43.2
Spain	4.4	5.89	14.0	21.8
Germany	4.4	13.95	12.3	54.3
Switzerland	4.4	17.96	11.9	108.2
Lithuania	4.6	7.74	20.4	22.0
Norway	5.0	17.88	11.3	79.8
Portugal	5.3	7.08	15.8	26.6
Austria	5.3	10.37	14.0	40.4
Greece	6.2	3.38	12.5	66.9

**Table 3 ijerph-19-16733-t003:** Percentage of residents covered by primary health care and satisfied with the quality and availability of the health care in selected OECD countries in 2019 [23].

Country	% of the Population Covered by Primary Health Care	% of the Population Satisfied with the Availability and Quality of Healthcare
Mexico	80.6	48
United States	89.8	83
Poland	93.4	26
Hungary	94	62
Slovak Republic	94.6	58
Colombia	94.7	47
Estonia	95	61
Chile	95.7	39
Belgium	98.6	92
Lithuania	98.7	51
Turkey	98.8	62
Austria	99.9	86
France	99.9	71
Netherlands	99.9	92
Australia	100	83
Canada	100	78
Czech Republic	100	75
Denmark	100	89
Finland	100	85
Germany	100	85
Greece	100	38
Iceland	100	81
Ireland	100	66
Israel	100	72
Italy	100	61
Japan	100	73
Korea	100	71
Latvia	100	no data
Luxembourg	100	85
New Zealand	100	77
Norway	100	93
Portugal	100	67
Slovenia	100	85
Spain	100	70
Sweden	100	82
Switzerland	100	91
United Kingdom	100	75

**Table 4 ijerph-19-16733-t004:** Pattern coordinate values after standardization.

Pattern Coordinates values	X1	X2	X3	X4	X5	X6	X7	X8	X9	X10
	2.75	2.17	2.38	2.45	3.46	3.46	−1.71	3.25	0.47	1.32

Note: own calculations.

**Table 5 ijerph-19-16733-t005:** Distance from the pattern (d_i0_) and synthetic measure of development (SMR_i_).

Country	d_i0_	SMR_i_
Australia	19.45	0.41
Austria	18.52	0.44
Belgium	20.47	0.38
Canada	25.08	0.24
Chile	31.42	0.05
Colombia	31.73	0.04
Czech Republic	21.26	0.36
Denmark	18.20	0.46
Estonia	27.65	0.16
Finland	20.21	0.39
France	21.28	0.36
Germany	15.95	0.52
Greece	24.32	0.26
Hungary	25.41	0.23
Iceland	20.21	0.39
Ireland	22.76	0.31
Israel	27.96	0.15
Italy	24.62	0.25
Japan	19.05	0.42
Korea	22.17	0.33
Latvia	28.53	0.14
Lithuania	23.82	0.28
Luxembourg	26.24	0.21
Mexico	36.77	−0.11
Netherlands	19.30	0.41
New Zealand	23.61	0.28
Norway	15.08	0.54
Poland	31.37	0.05
Portugal	23.49	0.29
Slovak Republic	25.81	0.22
Slovenia	21.16	0.36
Spain	24.61	0.25
Sweden	19.81	0.40
Switzerland	17.96	0.46
Turkey	30.28	0.08
United Kingdom	24.06	0.27
United States	21.35	0.35

Abbreviations: d_i0,_ distance from the pattern; SMRi, synthetic measure of development.

**Table 6 ijerph-19-16733-t006:** Ranking of health care systems in selected OECD countries.

Place	Country
1	Norway
2	Germany
3	Switzerland
4	Denmark
5	Austria
6	Japan
7	Netherlands
8	Australia
9	Sweden
10	Finland
11	Iceland
12	Belgium
13	Slovenia
14	Czech Republic
15	France
16	United States
17	Korea
18	Ireland
19	Portugal
20	New Zealand
21	Lithuania
22	United Kingdom
23	Greece
24	Spain
25	Italy
26	Canada
27	Hungary
28	Slovak Republic
29	Luxembourg
30	Estonia
31	Israel
32	Latvia
33	Turkey
34	Poland
35	Chile
36	Colombia
37	Mexico

## Data Availability

The data presented in this study are available on request from the corresponding author.

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
