# Peer review of "Assessment of the Health Care System in Poland and Other OECD Countries Using the Hellwig Method"

_ijerph, 2022, doi:10.3390/ijerph192416733_

Round 1
Reviewer 1 Report
This manuscript analyzes selected indicators of public health and the comparison the Polish health care system with systems of other countries. The authors are using secondary data from OECD databases. The authors presents to evaluate the Polish health care system against other Europe countries based on a set of indicators related to funding, medical and nursing staff, as well as the availability and quality of services provided. This manuscript raise a very important issue. In the reviewer's opinion, the article should be published. However, there are some areas where the manuscript needs revision. Below are some suggestions to improve the article:
1. The authors write: "The article presents a picture of the Polish health care system against the background of systems of other European countries" (Abstract), "The aim of the article is to analyze the Polish health care system and comparing it with other European systems based on selected indicators and indicating directions of changes.” (Abstract), “Therefore, the aim of this article is to evaluate the Polish health care system against other European countries based on a set of indicators related to funding, medical and nursing staff, as well as the availability and quality of services provided [ 9].” (Introduction), while the article analyzes OECD member countries. Not all OECD countries are in Europe. Please check the text so that the same information is given. The current one can be misleading.
2. Please write research hypothesis and consider ending the introduction by the aim statement. Any other argumentations after stating the aim of the study, may confuse the reader.
3. Please do not use subsections in the introduction and add information on health systems in other countries. Please consider reorganizing the introduction.
4. The reporting of the methods section, in the view of the reviewer, needs to be improved. This section is very long. The reporting of the methods section begins with subsection 2.1. This subsection should be deleted and included in the results. I propose to start section 2 with a few sentences (without second level of numbering). Please consider reorganizing to ease this section understanding.
5. Brazil, China, India, Russia and South Africa are not members of the OECD. They are listed as "Non-OECD economies." Why countries with this group are these considered? There are more countries in this group, so why are these countries excluded? The article presents only OECD member countries. (page 8, lines 183-184)
6. In the methods section, please explain the abbreviations from table 5 and table 6 and what program the authors used for the analysis.
7. Please specify the titles of the tables and figure, currently it is not known which year they refer to (Table 1, 2 and 3; Figure 1) and in what units the values are given (Table 1). I propose to standardize the presentation of numbers in tables, i.e. give the same number of decimal places.
8. Please provide the limitations of the study under the discussion section or under the conclusion section.
9. More generally, consider rephrasing some sentences for better understanding. For example, the authors write: “However, depending on the level of funding, the number of medical staff or their availability, there are significant discrepancies in the health care systems of different countries.” Availability – people or services? (page 1, line 13) or "An important element in the analysis and evaluation of health systems is their accessibility." Availability of what? systems? health services? (page 8, lines 172-173) or the authors write "In 2019, the average% of GDP allocated to financing health care was 8.86%" Which group of countries does this sentence refer to? Countries under consideration or others? (page 4, line 119)
Author Response
Thank you very much for your review and valuable comments about the article.
Most of the suggestions were followed up and the article improved.
- The authors write: "The article presents a picture of the Polish health care system against the background of systems of other European countries" (Abstract), "The aim of the article is to analyze the Polish health care system and comparing it with other European systems based on selected indicators and indicating directions of changes.” (Abstract), “Therefore, the aim of this article is to evaluate the Polish health care system against other European countries based on a set of indicators related to funding, medical and nursing staff, as well as the availability and quality of services provided [ 9].” (Introduction), while the article analyzes OECD member countries. Not all OECD countries are in Europe. Please check the text so that the same information is given. The current one can be misleading.
All comments have been entered.
- Please write research hypothesis and consider ending the introduction by the aim statement. Any other argumentations after stating the aim of the study, may confuse the reader.
Hypotheses were formulated (p. 3, lines 114-118).
The introduction ended with the definition of the goal (p. 3, lines 105-106).
- Please do not use subsections in the introduction and add information on health systems in other countries. Please consider reorganizing the introduction.
Introduction has been reorganized. Section 1.1 removed.
- The reporting of the methods section, in the view of the reviewer, needs to be improved. This section is very long. The reporting of the methods section begins with subsection 2.1. This subsection should be deleted and included in the results. I propose to start section 2 with a few sentences (without second level of numbering). Please consider reorganizing to ease this section understanding.
Section 2.1 has been moved to Chapter 3.
- Brazil, China, India, Russia and South Africa are not members of the OECD. They are listed as "Non-OECD economies." Why countries with this group are these considered? There are more countries in this group, so why are these countries excluded? The article presents only OECD member countries. (page 8, lines 183-184) –
The sentence about countries that have been rejected for analysis purposes has been deleted.
- In the methods section, please explain the abbreviations from table 5 and table 6 and what program the authors used for the analysis.
An explanation of abbreviations has been added to table 5 (p. 10, lines 239-240).
- Please specify the titles of the tables and figure, currently it is not known which year they refer to (Table 1, 2 and 3; Figure 1) and in what units the values are given (Table 1). I propose to standardize the presentation of numbers in tables, i.e. give the same number of decimal places.
Table 1 adds per capita (USD PPP) for each indicator and specifies values ​​up to 2 places after the decimal point. Diagram 1 changed (p. 6).
- Please provide the limitations of the study under the discussion section or under the conclusion section.
The limitations of the study were added in the conclusion section (p.13, lines 337-339).
- More generally, consider rephrasing some sentences for better understanding. For example, the authors write: “However, depending on the level of funding, the number of medical staff or their availability, there are significant discrepancies in the health care systems of different countries.” Availability – people or services? (page 1, line 13) or "An important element in the analysis and evaluation of health systems is their accessibility." Availability of what? systems? health services? (page 8, lines 172-173) or the authors write "In 2019, the average% of GDP allocated to financing health care was 8.86%" Which group of countries does this sentence refer to? Countries under consideration or others? (page 4, line 119)
The comments indicated in point 9 have been specified in detail. The indicated value of the average GDP was calculated based on the results of all the analyzed countries (p. 5, lines 167-168).

Reviewer 2 Report
Thank you for the opportunity to read and review the article "Selected Indicators of the health care system in Poland and other OECD countries: A comparative study using the Hellwig". The subject of the article is important, therefore any scientific study that brings us closer to this issue is desirable. The authors attempted to create a ranking of healthcare systems in selected OECD countries based on the Hellwig method. The obtained results allowed e.g. identification of benchmarks for countries interested in improving their health systems. However, the article requires many significant changes and additions.
General comments:
1. The title of the article needs to be changed so that it refers to the purpose of the article.
2. The article lacks a proper literature review. Has anyone already used a similar approach in comparing health systems? What methods did he use then?
3. There is no justification for why the Hellwig method was chosen. What is its advantage over other methods, e.g. TOPSiS? What other studies has it been used in? Examples of the application of this method or its modifications can be found e.g. in MDPI journals.
4. One gets the impression that most of the conclusions in the Discussion could be drawn without calculating the Hellwig measure.
Detailed comments:
1. The Introduction is missing a paragraph with the structure of the article.
2. The authors did not justify the choice of indicators (the results of the literature review mentioned in lines 196-197 are not included).
3. Tables 1-3 and Figure 1 should be placed in the Appendix.
4. The authors did not justify the adoption of equal weights.
5. The authors did not check the degree of correlation between the indicators.
6. What values can the Hellwig measure take?
7. Reference is missing where to find the report (line 200).
8. Please give all numbers accurate to 3 decimal places.
9. There is no explanation in Table 5 di0 and SMRi. What is the difference between qi and SMRi?
10. Numbering of some patterns is missing.
11. There are typos in the article, e.g. correction (line 229), mary (line 204).
Taking into account the above comments should improve the quality of the article and bring it closer to publication. Good luck!
Author Response
Thank you very much for your review and valuable comments about the article.
Most of the suggestions were followed up and the article improved.
General comments (1-4):
The title of the article needs to be changed so that it refers to the purpose of the article. The article lacks a proper literature review. Has anyone already used a similar approach in comparing health systems? What methods did he use then? There is no justification for why the Hellwig method was chosen. What is its advantage over other methods, e.g. TOPSiS? What other studies has it been used in? Examples of the application of this method or its modifications can be found e.g. in MDPI journals. One gets the impression that most of the conclusions in the Discussion could be drawn without calculating the Hellwig measure.
As a result of verifying the purpose of the publication, the title was changed to Assessment of the health care system in Poland and other OECD countries using the Hellwig method. No publication was found that would use the Hellwig method to compare health care systems. Hence, it was considered an interesting approach. Therefore, they are used in this publication.
Detailed comments:
- The Introduction is missing a paragraph with the structure of the article.
Added (p.3, lines 106-108).
- The authors did not justify the choice of indicators (the results of the literature review mentioned in lines 196-197 are not included).
On the basis of a literature review, selected problems concerning the Polish health care system were diagnosed, to which appropriate indicators were selected.
- Tables 1-3 and Figure 1 should be placed in the Appendix.
From the point of view of logic and readability of the argument, Table 1-3 and Figure 1 have been left in the text.
- The authors did not justify the adoption of equal weights.
Equal weight values ​​were assumed: 1 for all indicators.
- The authors did not check the degree of correlation between the indicators.
Due to the choice of the Hellwig method aimed at creating a ranking of countries, correlations between indicators were not verified.
- What values can the Hellwig measure take?
The values ​​of the synthetic measure in the Hellwig method are usually in the range [0;1]. (p. 4).
- Reference is missing where to find the report (line 200).
Source completed (p.3, line 120).
- Please give all numbers accurate to 3 decimal places.
The publication assumes that all numerical values ​​will be presented with an accuracy of one or two places after the decimal point.
- There is no explanation in Table 5 di0 and SMRi. What is the difference between qi and SMRi?
An explanation of the abbreviations used has been added to Table 5 (p. 10, line 236).
- Numbering of some patterns is missing.
Patterns have been numbered (p.4)
- There are typos in the article, e.g. correction (line 229), mary (line 204).
Fixed typos.

Round 2
Reviewer 2 Report
The article has been corrected and meets the publication requirements.